# MANIFOLD REGULARIZATION WITH GANS FOR SEMI-SUPERVISED LEARNING

## ABSTRACT

Generative Adversarial Networks are powerful generative models that can model the manifold of natural images. We leverage this property to perform manifold regularization by approximating a variant of the Laplacian norm using a Monte Carlo approximation that is easily computed with the GAN. When incorporated into the semi-supervised feature-matching GAN we achieve state-of-the-art results for semi-supervised learning on CIFAR-10 benchmarks when few labels are used, with a method that is significantly easier to implement than competing methods. We find that manifold regularization improves the quality of generated images, and is affected by the quality of the GAN used to approximate the regularizer.

## 1 INTRODUCTION

Deep neural network classifiers typically require large labeled datasets to obtain high predictive performance. Obtaining such a dataset could be time and cost prohibitive especially for applications where careful expert labeling is required, for instance, in healthcare and medicine. Semi-supervised learning algorithms that enable models to be learned from a small amount of labeled data augmented with (large amounts of) unlabeled data have the potential to vastly reduce this labeling burden.

Fundamentally, semi-supervised learning requires assumptions relating the distribution of the data $\mathcal{P}_x$ (which can be derived from the unlabeled data) to the classification task (Chapelle et al., 2010). For instance, the classic manifold regularization framework (Belkin et al., 2006) for semi-supervised learning makes the assumption that that the data lie on a low-dimensional manifold $\mathcal{M}$ and moreover that a classifier $f$ is smooth on this manifold, so nearby points on the manifold are assigned similar labels. Algorithms based on this framework enforce a classifier's invariance to local perturbations on the manifold by penalizing its Laplacian norm $\|f\|_L^2 = \int_{x \in M} \|\nabla_{\mathcal{M}} f(x)\|^2 \, d\mathcal{P}_X(x)$. More generally, regularization terms penalizing classifier gradients in regions of high data density have also been proposed (Bousquet et al., 2003).

Recently, generative adversarial networks (GANs) have been used for semi-supervised learning, where they are competitive with state-of-the-art methods for semi-supervised image classification (Dai et al., 2017; Kumar et al., 2017; Qi et al., 2018). GAN-based semi-supervised learning methods typically build upon the formulation in (Salimans et al., 2016), where the discriminator is extended to determine the specific class of an image or whether it is generated; by contrast, the original GAN's discriminator is only expected to determine whether an image is real or generated. Another key application for GANs is image synthesis, where they have been shown to model the image manifold well (Zhu et al., 2016). Recent work (Kumar et al., 2017; Qi et al., 2018) has used this property of GANs to enforce discriminator invariance on the image manifold, resulting in improved accuracy on semi-supervised image classification.

In this work, we leverage the ability of GANs to model the image manifold to efficiently approximate the Laplacian norm and related regularization terms through Monte-Carlo integration. We show that classifiers (with varying network architectures) regularized with our method outperform baselines on the SVHN and CIFAR-10 benchmark datasets with a method that is *significantly simpler to implement than competing methods*. In particular, when applied to the semi-supervised feature-matching GAN (Salimans et al., 2016), our method achieves *state-of-the-art performance amongst GAN-based methods*, and is highly competitive with other non-GAN approaches *especially when the number of labeled examples is small*. We show that manifold regularization improves the quality

of generated images as measured by Inception and FID scores when applied to the semi-supervised feature-matching GAN, thus *linking manifold regularization to recent work on gradient penalties for stabilizing GAN training* (Roth et al., 2017; Gulrajani et al., 2017; Mescheder et al., 2017). We also found that generator quality (as measured by the quality of generated images) influences the benefit provided by our manifold regularization strategy in that using a better quality generator results in larger improvements in classification performance over a supervised baseline.

## 2 RELATED WORK

There have been several works adapting GANs for semi-supervised learning. One approach is to change the standard binary discriminator of a standard GAN to predict class labels of labeled examples, while enforcing the constraint that generated data should result in uncertain classifier predictions (Springenberg, 2016). The related approach of Salimans et al. (2016) also uses the discriminator of the GAN as the final classifier, but instead modifies it to predict $K + 1$ probabilities ($K$ real classes and the generated class). This approach performs well when combined with a feature matching loss for the generator. The work of Li et al. (2017) introduces an additional classifier as well as uses a conditional generator instead of adapting the discriminator to overcome limitations with the two-player formulation of standard GANs in the context of semi-supervised learning.

The idea of encouraging local invariances dates back to the TangentProp algorithm (Simard et al., 1991) where manifold gradients at input data points are estimated using explicit transformations of the data that keep it on the manifold, for example small rotations and translations. Since then other approaches have tried to estimate these tangent directions in different ways. High order contractive autoencoders were used by Rifai et al. (2011b) to capture the structure of the manifold; this representation learning algorithm was then used to encourage a classifier to be insensitive to local direction changes along the manifold (Rifai et al., 2011a). This approach was recently revisited in the context of GANs (Kumar et al., 2017), where the tangent space to the data manifold is estimated using GANs with an encoder in order to inject invariance into the classifier. In addition, this work also explored the use of an additional ambient regularization term which promotes invariance of the discriminator to perturbations on training images along all directions in the data space. The proposed method is competitive with the state-of-the-art GAN method of (Dai et al., 2017), which argues that a generator that generates images that are in the complement of the training data distribution is necessary for good semi-supervised learning performance. Most recently (Qi et al., 2018) proposed the use of a local GAN which attempts to model the local manifold geometry around data points without the need for an encoder. The local GAN is then used to approximate the Laplacian norm for semi-supervised learning, and is shown to enable state-of-the-art classification results.

Aside from GAN-based approaches, Virtual Adversarial Training (VAT) (Miyato et al., 2017), which is based on constraining model predictions to be consistent to local perturbation, has also achieved state-of-the-art performance on benchmarks. Specifically, VAT smooths predictions of the classifier over adversarial examples centered around labeled and unlabeled examples. (Park et al., 2018) similarly introduces adversarial pruning of the neural network to regularize the training. This method gives state-of-the-art results, when combined with VAT.

Other recent works are based on the self-training paradigm (Chapelle et al., 2010). Such methods label the unlabeled data using classifiers trained on the labeled data, and then use this expanded labeled dataset to train a final classifier. Recent progress has been due to clever use of ensembling to produce better predictions on the unlabeled data: instead of simply using predictions of a model trained on the labeled data, (Laine & Aila, 2017) ensembled model predictions under various perturbations or at different time steps. In follow-up work, the Mean Teacher method (Tarvainen & Valpola, 2017) averages model weights instead at different time steps using exponential moving averages, and achieves state-of-the-art performance on benchmark image datasets. Most recently, the work of (Luo et al., 2018) proposes a consistency loss that considers perturbations not only on a datapoint but also on its neighbors.

## 3 MANIFOLD REGULARIZATION

We present an approach to approximate any density-based regularization term of the form $\Omega(f) = \int_{x \in \mathcal{M}} L(f) \mathrm{d}\chi(\mathcal{P}_X)$ (Bousquet et al., 2003) with $L$ denoting a measure of smoothness of the clas-

sifier function $f$ and $\chi$ a strictly-increasing function. This class of regularizers enforce classifier smoothness in regions of high data density and includes the Laplacian norm with $L(f) = \|\nabla_{\mathcal{M}} f\|^2$ and $\chi$ the identity function. We focus on the following variant of the Laplacian norm

$$\Omega(f) = \int_{x \in \mathcal{M}} \|\nabla_{\mathcal{M}} f\|_F \, d\mathcal{P}_X$$

with $L(f) = \|\nabla_{\mathcal{M}} f\|_F$ in this work and show how it can be approximated efficiently [1]. Our approach relies on two commonly held assumptions about GANs:

1. GANs can model the distribution over images (Radford et al., 2016), such that samples from the GAN are distributed approximately as $\mathcal{P}_X(x)$, the marginal distribution over images $x$.
2. GANs learn the image manifold (Radford et al., 2016; Zhu et al., 2016). Specifically, we assume that the generator $g$ learns a mapping from the low-dimensional latent space with coordinates $z$ to the image manifold embedded in a higher-dimensional space, enabling us to compute gradients on the manifold by taking derivatives with respect to $z$ (Kumar et al., 2017; Shao et al., 2017) .

With these assumptions, we may approximate $\Omega(f)$ as follows, where we list the relevant assumption above each approximation step

$$\Omega(f) = \int_{x \in \mathcal{M}} \|\nabla_{\mathcal{M}} f\|_F \, d\mathcal{P} \stackrel{(1)}{\approx} \frac{1}{n} \sum_{i=1}^{n} \left\| \nabla_{\mathcal{M}} f(g(z^{(i)})) \right\|_F \stackrel{(2)}{\approx} \frac{1}{n} \sum_{i=1}^{n} \left\| J_z f(g(z^{(i)})) \right\|_F .$$

Here, $J_z$ denotes the Jacobian matrix of partial derivatives of classifier outputs $f$ with respect to latent generator variables $z$ [2]. Computing gradients of $\Omega(f)$ during model learning is computationally prohibitive for deep neural networks as it requires computing the Hessian of a model with large numbers of parameters [3]. We hence used stochastic finite differences to approximate the gradient term for computational efficiency.

**Issues with manifold gradients:** To motivate the specific approximation we used, we first illustrate several issues with manifold gradients as estimated with a GAN when considering the obvious candidate approximation $\left\| J_z f(g(z^{(i)})) \right\|_F \approx \left\| f(g(z^{(i)})) - f(g(z^{(i)} + \delta)) \right\|_F, \delta \sim \mathcal{N}(0, \sigma^2 I)$ (Figure 1), using the Two Circles dataset and MNIST. The Two Circles dataset is an example of data lying on disjoint manifolds. In this case, even though the GAN is able to accurately model the data distribution (Figure 1a, left), we see several instances where the manifold gradients as per the GAN are extremely noisy (Figure 1a, center) with large magnitudes. If data from the inner and outer circle were to belong to different classes, enforcing classifier smoothness at those points in the manifold with these large noisy gradients could result in the classifier predicting similar values for both circles (as $g(z^{(i)})$ and $g(z^{(i)} + \delta) \approx g(z^{(i)}) + J_z(g(z^{(i)}))\delta$ lie on different circles), causing erroneous classifications. At the other extreme, there are points on the manifold where the estimated gradient has such small magnitude such that the regularizer has minimal smoothing effect. These issues are also evident on the MNIST dataset (Figure 1b), where we directly show how $g(z^{(i)})$ and $g(z^{(i)} + \delta)$ can lie on different manifolds (red rectangle) or be virtually identical (blue rectangle), resulting in over-smoothing and under-smoothing respectively of the classifier function. We note that the analysis of Chen et al. (2018) suggests that the issue of large noisy gradients that we described may be inherent to generative latent variable models, as such models may place points from disjoint manifolds in the data space nearby in the latent space (see Figure A4 for an example).

**Our improved gradient approximation:** In light of these issues arising from the magnitude of manifold gradients, we used the following approximation that takes a step of tunable size $\epsilon$ in the direction of the manifold gradient, thus ignoring the magnitude of the gradient while enforcing smoothness in its direction

$$\Omega(f) \approx \frac{1}{n} \sum_{i=1}^{n} \left\| f\left(g\left(z^{(i)}\right)\right) - f\left(g\left(z^{(i)}\right) + \epsilon \bar{r}\left(z^{(i)}\right)\right) \right\|_F$$

---

[1]In our early experiments we used the regular Laplacian norm but found that this variant worked better in a wider range of settings; results using the Laplacian norm are included in the Appendix.

[2]In our experiments, we defined $f$ to be the logits of the softmax output layer instead of the resultant normalized probabilities as we found it gave better performance.

[3]In fact, for multi-class classifiers $f$, we need to compute a Hessian *tensor* – one matrix for each component (class output) of $f$, which quickly becomes impractical even with moderate numbers of classes.

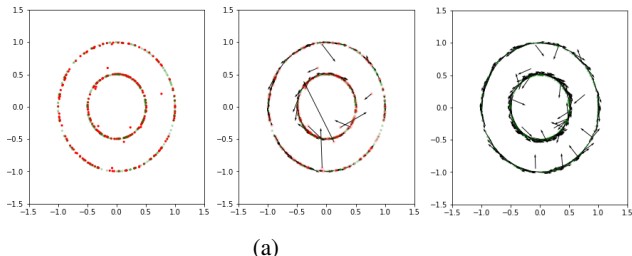
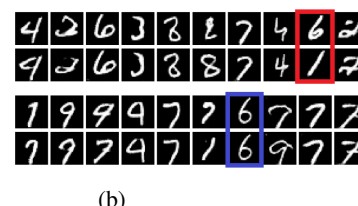

(a)

(b)

Figure 1: Issues with GAN-derived manifold gradients. (a) Left: Samples (green dots) and generated samples (red dots) from a GAN trained on the two circles dataset. Middle: Manifold gradients from the trained GAN. Right: Manifold gradients normalized to unit norm. The GAN approximates the data manifold well in this toy example and normalizing gradients mitigates issues with noisy gradients. (b) Effect of perturbations on the latent code of a GAN trained on MNIST. Each pair of rows shows the generated example from the latent code (top) and the generated example from the perturbed latent code (bottom). Random perturbations of equal norm in the latent space can have almost no effect (blue box) or a large effect (red box) on generated examples.

Here $r(z) = g(z + \eta\,\bar{\delta}) - g(z^{(i)}), \delta \sim \mathcal{N}(0, I)$ is an approximation of the manifold gradient at $z$ with tunable step size $\eta$, and $\bar{v} = \frac{v}{\|v\|}$ denotes a unit vector. Note that the stochastic finite difference approximates a directional derivative in the randomly chosen direction $\delta$: $f(g(z) + \epsilon\bar{r}) - f(g(z)) \approx \epsilon J_f . J_g \frac{\bar{\delta}}{\|J_g\bar{\delta}\|}$. To gain further intuition, we can rewrite the expression $g(z) + \epsilon\bar{r}(z)$ as $g(z)(1 - \frac{\epsilon}{\|r(z)\|}) + g(z + \eta\,\bar{\delta})\frac{\epsilon}{\|r(z)\|}$ to see that a large manifold gradient (large $\|r(z)\|$) will cause $f$ to be evaluated closer to $g(z)$ instead of $g(z + \eta\,\bar{\delta})$, which is the desired behavior.

**Advantages of our approach**: We wish to highlight that our approach only relies on training a standard GAN. In contrast to the approach of Kumar et al. (2017), we do not explicitly enforce classifier smoothness on input data points, allowing us to avoid the added complexity of learning an encoder network to determine the underlying manifold coordinates for a data sample, as well as other tricks required to estimate tangent directions. Our approach leverages the GAN's ability to interpolate smoothly between input images to enforce classifier smoothness on the entirety of the manifold and not just at the provided data points. We note that Qi et al. (2018) proposed an alternate and elegant solution to the issues we identified by learning a local GAN instead at input data points, but at the price of training a GAN with a more complex local generator.

## 4 EXPERIMENTS

### 4.1 EXPERIMENTAL SETUP

Our experimental setup follows (Tarvainen & Valpola, 2017) – we separated 10% of the training data on CIFAR-10 into a validation set, and did similarly for SVHN. We designed our algorithms and chose our hyperparameters based on this validation set. We perform semi-supervised training using a small fraction of the labeled training data containing an equal number of examples from each class; the remaining training images are used as unlabeled data. Note that the classifier we used for evaluation had weights that were the exponential moving average of classifier weights obtained during training (this technique was also used in (Salimans et al., 2016)). We report the error rates on the test set for models which performed best on the validation set. Details of hyperparameters and network architectures we used in our experiments can be found in the Appendix.

### 4.2 INCORPORATING MANIFOLD REGULARIZATION INTO SEMI-SUPERVISED GANS

We first evaluated our regularization method when incorporated into the semi-supervised GAN framework. We reproduced the semi-supervised feature-matching GAN of (Salimans et al., 2016) and added our manifold regularizer to the model. The final loss function of the discriminator is:

$$L = L_{supervised} + L_{unsupervised} + \gamma_m\,\Omega_{manifold}, \text{ where}$$

$$
\begin{aligned}
L_{unsupervised} &= -\mathbb{E}_{x \sim p_{data}(x)} \left[ \log \left[ 1 - p_f(y = K+1|x) \right] \right] - \mathbb{E}_{x \sim g} \left[ \log \left[ p_f(y = K+1|x) \right] \right] \\
L_{supervised} &= -\mathbb{E}_{x,y \sim p_{data}(x,y)} \left[ \log p_f(y|x, y < K+1) \right] \\
\Omega_{manifold} &= \mathbb{E}_{z \sim U(z), \delta \sim N(\delta)} \left\| f\left(g\left(z\right)\right) - f\left(g\left(z\right) + \epsilon \bar{r}\right) \right\|_2
\end{aligned}
$$

and we used the feature matching loss for our generator, $\left\| \mathbb{E}_{x \sim p_{data}} h(x) - \mathbb{E}_{z \sim p_z(z)} h(g(z)) \right\|$. Here, $h(x)$ denotes activations on an intermediate layer of the discriminator. We also checked if the anisotropic regularization along manifold directions in the data space that we used provides additional benefits over simple ambient regularization in the data space. Specifically, we evaluated the ambient regularizer $\lambda \, \mathbb{E}_{x \sim p_d(x)} \| J_x f \|$, as proposed in (Kumar et al., 2017) that we similarly approximate using a stochastic finite difference as shown below: $L = L_{supervised} + L_{unsupervised} + \gamma_a \Omega_{ambient}$, where

$$
\Omega_{ambient} = \mathbb{E}_{\delta \sim N(\delta)} \left\| f\left(x\right) - f\left(x + \epsilon \, \bar{\delta}\right) \right\|_2.
$$

Table 1: Error rate on CIFAR-10 averaged over 3 runs with different random seeds. Results including data augmentation (4x4 random translations and random horizontal flips similar to previous works). * marks baselines we ran; the remainder are taken from the relevant papers.

| CIFAR-10 | 1000 labels (2%) 50000 images | 2000 labels (4%) 50000 images | 4000 labels (8%) 50000 images |
|---|---|---|---|
| Mean Teacher (Tarvainen & Valpola, 2017) | $21.55 \pm 1.48$ | $15.73 \pm 0.31$ | $12.31 \pm 0.28$ |
| $\Pi$ (Laine & Aila, 2017) | $31.65 \pm 1.20$ | $17.57 \pm 0.44$ | $12.36 \pm 0.31$ |
| TempEns (Laine & Aila, 2017) | $23.31 \pm 1.01$ | $15.64 \pm 0.39$ | $12.16 \pm 0.31$ |
| VAT (Miyato et al., 2017) | 18.34* | 14.20* | 11.36 |
| VAT + EntMin (Miyato et al., 2017) | $17.13 \pm 0.91$* | $14.25 \pm 0.74$* | 10.55 |
| VAdD (QE) (Park et al., 2018) | | | 11.32 |
| VAdD (QE) + VAT (Park et al., 2018) | | | **9.22** |
| Pi + SNTG (Luo et al., 2018) | $21.23 \pm 1.27$ | $14.65 \pm 0.31$ | $11.00 \pm 0.13$ |
| TempEns+SNTG (Luo et al., 2018) | $18.41 \pm 0.52$ | $13.64 \pm 0.32$ | $10.93 \pm 0.14$ |
| VAT + Ent + SNTG (Luo et al., 2018) | | | 9.89 |
| **Manifold (ours)** (4 runs) | **$13.71 \pm 0.24$** | **$12.99 \pm 0.49$** | $11.79 \pm 0.25$ |

Table 2: Error rate on CIFAR-10 averaged over 4 runs with different random seeds. Results were obtained without data augmentation.

| CIFAR-10 | 1000 labels (2%) 50000 images | 2000 labels (4%) 50000 images | 4000 labels (8%) 50000 images |
|---|---|---|---|
| $\Pi$ model (Laine & Aila, 2017) | $32.18 \pm 1.33$ | $23.92 \pm 1.03$ | $16.55 \pm 0.29$ |
| Mean Teacher (Tarvainen & Valpola, 2017) | $30.62 \pm 1.13$ | $23.14 \pm 0.46$ | $17.74 \pm 0.30$ |
| VAT (large) (Miyato et al., 2017) | 22.18* | 18.32* | 14.18 |
| VAT+EntMin (large) (Miyato et al., 2017) | $19.89 \pm 0.89$* | $15.78 \pm 0.36$* | 13.15 |
| $\Pi$+SNTG (Luo et al., 2018) | | | $13.62 \pm 0.17$ |
| VAT+Ent+SNTG (Luo et al., 2018) | | | **$12.49 \pm 0.36$** |
| Improved GAN (Salimans et al., 2016) | $21.83 \pm 2.01$ | $19.61 \pm 2.09$ | $18.63 \pm 2.32$ |
| Improved GAN + SNTG (Luo et al., 2018) | | | 14.93 |
| Improved Semi-GAN (Kumar et al., 2017) | $19.52 \pm 1.50$ | | $16.20 \pm 1.6$ |
| ALI (Dumoulin et al., 2017) | $19.98 \pm 0.89$ | $19.09 \pm 0.44$ | $17.99 \pm 1.62$ |
| Bad GAN (Dai et al., 2017) | | | $14.41 \pm 0.30$ |
| Local GAN (Qi et al., 2018) | $17.44 \pm 0.25$ | | $14.23 \pm 0.27$ |
| Improved GAN (ours) | $17.50 \pm 0.34$ | $16.80 \pm 0.07$ | $15.5 \pm 0.35$ |
| Ambient (ours) | $16.98 \pm 0.36$ | $15.99 \pm 0.14$ | $14.75 \pm 0.37$ |
| Manifold reg. unnormalized (ours) | 18.09 | 15.8 | $14.40 \pm 0.21$ |
| **Manifold (ours)** | **$16.37 \pm 0.42$** | **$15.25 \pm 0.35$** | $14.34 \pm 0.17$ |

We present results on CIFAR-10 (Krizhevsky, 2009) in in Tables 1 and 2, and results on SVHN (Netzer et al., 2011) in Table 3. We first note that our implementation of the feature-matching GAN with weight normalization (Salimans & Kingma, 2016) (Improved GAN) (Salimans et al., 2016) significantly outperforms the original (and many other recent methods) after we tuned training hyperparameters, illustrating the sensitivity of semi-supervised GANs to hyperparameter settings. Adding manifold regularization to the feature-matching GAN further improves performance, achieving state-of-the-art results amongst all GAN-based methods, as well as being highly competitive with other non-GAN-based methods, especially when less labeled data is used. We also observe

that while simple (isotropic) ambient regularization provides some benefit, our (anisotropic) manifold regularization term provides additional performance gains.

Our results are consistent with recent work in semi-supervised learning and more generally regularization of neural networks. While studies have shown that promoting classifier robustness against local perturbations is effective for semi-supervised learning (Laine & Aila, 2017; Tarvainen & Valpola, 2017), other recent work suggests that it is difficult to achieve local isotropy by enforcing invariance to random perturbations independent of the inputs (which is what the simple ambient regularizer does) in highly non-linear models (Szegedy et al., 2014), so data-dependent perturbations should be used instead. One possibility is to enforce invariance to perturbations along adversarial directions of the classifier (Miyato et al., 2017). Our approach instead enforces invariance to perturbations on the data manifold as modelled by the GAN.

Table 3: Error rate on SVHN averaged over 4 runs with different random seeds. Results were obtained without data augmentation.

| SVHN | 500 labels(0.3%) 73257 images | 1000 labels(1.4%) 73257 images |
|---|---|---|
| Π model (Laine & Aila, 2017) | $7.01 \pm 0.29$ | $5.73 \pm 0.16$ |
| Mean Teacher (Tarvainen & Valpola, 2017) | $5.45 \pm 0.14$ | $5.21 \pm 0.21$ |
| VAT (large) (Miyato et al., 2017) | | 5.77 |
| VAT+EntMin(Large)(Miyato et al., 2017) | | 4.28 |
| VAT+Ent+SNTG (Luo et al., 2018) | | **4.02±0.20** |
| Π+SNTG (Luo et al., 2018) | | 4.22±0.16 |
| Improved GAN(Salimans et al., 2016) | $18.44 \pm 4.80$ | $8.11 \pm 1.3$ |
| Improved semi-GAN(Kumar et al., 2017) | **4.87 ±1.6** | $4.39 \pm 1.5$ |
| ALI (Dumoulin et al., 2017) | | $7.41 \pm 0.65$ |
| Triple-GAN (Li et al., 2017) | | $5.77 \pm 0.17$ |
| Bad GAN (Dai et al., 2017) | | $7.42 \pm 0.65$ |
| Local GAN (Qi et al., 2018) | $5.48 \pm 0.29$ | $4.73 \pm 0.29$ |
| Improved GAN (ours) | $6.13 \pm 0.41$ | $4.9 \pm 0.10$ |
| **Manifold regularization (ours)** | $5.67 \pm 0.11$ | $4.63 \pm 0.11$ |

## 4.3 Interaction of our regularizer with the generator

In the semi-supervised GAN framework, applying manifold regularization to the discriminator has the potential to affect the generator through the adversarial training procedure. We explored the effects of our regularization on the generator by evaluating the quality of images generated using the Inception (Salimans et al., 2016) and FID scores (Heusel et al., 2017) as shown in Table 4. We observe that adding manifold regularization yields significant improvements in image quality across both CIFAR-10 and SVHN datasets and with varying amounts of labeled data. Why might manifold regularization help? Here we briefly discuss two possible explanations for this improvement – gradient penalties and improved generator conditioning.

*Gradient penalties:* Our results are consistent with recent work suggesting that gradient penalties on the discriminator may be used to stabilize GAN training (Gulrajani et al., 2017; Roth et al., 2017; Mescheder et al., 2018b). Our regularization term is closely related to the proposed penalties in (Mescheder et al., 2018b). In particular, the proposed $R_2$ regularizer $\frac{\gamma}{2}\mathbb{E}_{x \sim P_g}[\ ||\nabla D_x(x)||^2]$ that penalizes the gradient of the discriminator $D$ with respect to the data space, over samples drawn from the generator distribution $P_g$, is almost identical to our proposed regularizer, except that we take gradients with respect to the generator's latent variables $z$, and we do not square the norm.

*Generator conditioning:* Recent work suggests that the condition number of the Jacobian of the generator (generator conditioning) is causally related to GAN performance as measured by Inception and FID scores in that well-conditioned generators perform better (Odena et al., 2018); we note that a similar stochastic finite difference regularizer was used in this work to encourage the spectrum of the generator Jacobian to lie in a specific range. If our manifold regularizer influenced generator conditioning, this could explain the improvements in GAN quality we observed. To test this hypothesis, we computed the mean log condition number (MLCN) of the generator with and without manifold regularization. Consistent with this hypothesis, we observed improved conditioning

with regularization (MLCN: 7.97 with regularization, 8.45 without); in this experiment GANs were trained with 1000 labeled training examples on the CIFAR-10.

Table 4: Comparison of Inception (Salimans et al., 2016) (higher is better) and FID scores (Heusel et al., 2017) (lower is better). Results shown are from 3 runs using different random seeds.

| CIFAR-10 | Inception Score |
|---|---|
| Unsupervised DCGANs (Radford et al., 2016) | $6.16 \pm 0.07$ |
| Supervised DCGANs (Radford et al., 2016) | 6.58 |
| Unsupervised GP-WGAN (Gulrajani et al., 2017) | $7.86 \pm 0.07$ |
| Supervised GP-WGAN (Gulrajani et al., 2017) | $8.42 \pm 0.1$ |
| Zero-centered GP (R2) (Mescheder et al., 2018a) | 6.20 |
| **1000 labels** | |
| Improved GAN | $6.28 \pm 0.01$ |
| + Manifold | $6.77 \pm 0.11$ |
| + Ambient | $6.90 \pm 0.01$ |
| **2000 labels** | |
| Improved GAN | $6.24 \pm 0.10$ |
| + Manifold | $6.69 \pm 0.05$ |
| + Ambient | $6.80 \pm 0.20$ |
| **4000 labels** | |
| Improved GAN | $6.24 \pm 0.13$ |
| + Manifold | $6.63 \pm 0.09$ |
| + Ambient | $6.87 \pm 0.10$ |

| CIFAR-10 | FID score |
|---|---|
| **1000 labels** | |
| Improved GAN | $38.59 \pm 0.18$ |
| + Manifold | $32.03 \pm 0.44$ |
| + Ambient | $27.58 \pm 0.69$ |
| **2000 labels** | |
| Improved GAN | $39.18 \pm 0.62$ |
| + Manifold | $33.09 \pm 0.65$ |
| + Ambient | $29.09 \pm 0.12$ |
| **4000 labels** | |
| Improved GAN | 39.23 |
| + Manifold | $33.84 \pm 1.08$ |
| + Ambient | $29.03 \pm 0.11$ |
| **SVHN** | |
| **1000 labels** | |
| Improved GAN | $86 \pm 12.98$ |
| + Manifold | $47.03 \pm 26.99$ |
| **500 labels** | |
| Improved GAN | $85.49 \pm 11.73$ |
| + Manifold | $38.65 \pm 7.33$ |

## 4.4 INCORPORATING MANIFOLD REGULARIZATION INTO CONVOLUTIONAL NEURAL NETS

We also explored the potential for our manifold regularization framework to improve performance of classifiers outside the semi-supervised GAN framework. We performed a series of experiments where we first trained a GAN to learn the marginal distribution of the data $\mathcal{P}_X$ and the data manifold $\mathcal{M}$, and subsequently used the trained GAN to regularize a separate neural network classifier. In this setup, we minimize the following loss where $V$ is the cross entropy loss on the labeled examples:

$$L = \frac{1}{n} \sum_n V(x^{(i)}, y^{(i)}, f) + \gamma_m \Omega(f).$$

Here unlabeled examples are implicitly used to regularize the classifier $f$ since they are used to train the GAN. This "decoupled" setting enables us to understand how the quality of the generator used to approximate the manifold regularizer affects classification accuracy *as there is no interaction between the generator and the classifier being regularized*.

We first verified that our method works on synthetic 2D datasets when the manifold is known exactly (Figure A2) and also when a GAN is used to learn both the data manifold and the data distribution on this manifold (Figure A3). In these experiments, we penalized a neural network classifier with our manifold regularizer as approximated with a consensus GAN (Mescheder et al., 2017). The classifier we used consists of 6 fully-connected layers with 384 neurons each. We observed that perfect classification could be achieved with only one labeled example per class even when the GAN does not perfectly learn the data distribution or manifold.

We then evaluated our manifold regularization method on a real image dataset (CIFAR-10), using a standard DCGAN to approximate the manifold regularizer and a 13 layer CNN as the classifier (see the Appendix for more details). We observe that manifold regularization is able to reduce classification error by 2-3% over the purely supervised baseline when different amounts of labeled data are used (Table 5 bottom; first two rows).

## 4.5 EFFECT OF GAN QUALITY

To quantify the importance of the first step of manifold learning, we also compared the performance of classifiers when regularized using GANs of differing quality (as assessed by the quality of gen-

erated images). We compared the following three generators: 1) a DCGAN which produces decent looking images (inception score of 6.68 and FID of 32.32 on CIFAR-10), 2) a DCGAN with a much lower inception score (inception score of 3.57 on CIFAR-10), and 3) a noise image generator where pixels are generated independently from uniform distributions in the range $[0, 255]$.

Table 5: Error rate on 4 runs of a CNN with manifold regularization using a separate GAN (DC-GAN). Results shown are obtained without data augmentation. Our models were not trained with ZCA whitening but results from other methods include ZCA whitening.

| CIFAR-10 | 1000 labels (2%) 50000 images | 2000 labels (4%) 50000 images | 4000 labels (8%) 50000 images |
|---|---|---|---|
| Supervised (Tarvainen & Valpola, 2017) | $48.38 \pm 1.07$ | $36.07 \pm 0.90$ | $24.47 \pm 0.50$ |
| $\Pi$ model (Laine & Aila, 2017) | $32.18 \pm 1.33$ | $23.92 \pm 1.03$ | $16.55 \pm 0.29$ |
| Mean Teacher (Tarvainen & Valpola, 2017) | $30.62 \pm 1.13$ | $23.14 \pm 0.46$ | $17.74 \pm 0.30$ |
| Supervised (ours) | $41.65 \pm 3.12$ | $32.46 \pm 0.52$ | $25.01 \pm 1.29$ |
| **+ manifold (IS: 6.68)** | $38.76 \pm 1.81$ | $29.44 \pm 0.45$ | $23.5 \pm 1.20$ |
| **+ manifold (IS: 3.57)** | $39.14 \pm 1.46$ | $32.84 \pm 2.13$ | $24.87 \pm 0.73$ |
| **+ manifold (noise)** | $66.99 \pm 5.23$ | $72.21 \pm 4.87$ | $67.79 \pm 4.30$ |

We observe that using GANs with better generators for manifold regularization resulted in lower classification errors (Table 5; last 3 rows). Our results also suggest that even GANs that generate lower quality images but nonetheless have captured some aspects of the image manifold are able to provide some performance benefit when used for manifold regularization. As a negative control, we observe that performance degrades relative to the supervised baseline when we compute the regularizer using randomly generated images.

### 4.6 Understanding our manifold regularization approximation method

Finally, we attempt to provide some intuition of how our manifold regularization approximation method works. Our method promotes classifier invariance between generated samples $g(z)$ and their corresponding perturbations $g(z) + \epsilon \bar{r}$ obtained by perturbing their latent code. In other words, the regularizer promotes invariance of the classifier specifically along directions on the data manifold. We show in Figure A8 some examples of generated samples and their corresponding perturbations. We see that even if some features such as the background color may change, there exists a global class consistency between the images as $\epsilon$ is varied. Hence, enforcing invariance in predictions along these directions will result in correct classifications provided $\epsilon$ is not too large (which would result in larger distortions and unrecognizable images). An intuitive explanation is that manifold regularization performs label propagation across "semantically similar" images by minimizing the manifold consistency cost $\Omega(f)$, so that an image of a red car gets the same label as an orange car.

## 5 Conclusion

GANs are powerful generative models that are able to model the distribution and manifold over natural images. We leverage these properties to perform manifold regularization by approximating a variant of the Laplacian norm using a Monte Carlo approximation that is easily computed with the GAN. We show that our regularization strategy consistently improves classification performance using unlabeled data on the CIFAR-10 and SVHN benchmarks, on several neural network architectures, and with varying amounts of labeled data. In particular, when incorporated into the feature-matching GAN of (Salimans et al., 2016), we achieve state-of-the-art results for semi-supervised image classification when few labels are used with a method that is significantly easier to implement than competing methods. We explored the interaction between our regularization and the generator in this framework and reveal a connection with gradient penalties for stabilizing GAN training. Using an experimental setup where we decoupled the GAN used for estimating the regularizer and the classifier, we further observed a positive correlation between generator image quality and prediction accuracy. Our work uses GANs in a novel way for semi-supervised classification, and we expect that our approach will be applicable to semi-supervised regression (Belkin & Niyogi, 2004; Lafferty & Wasserman, 2007; Moscovich et al., 2017) as well as unsupervised learning (Belkin et al., 2006).

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

APPENDICES

## A   ADDITIONAL RESULTS ON CIFAR-10

**Comparison of the finite difference methods:** We provide in this section additional results for the semi-supervised learning experiments with the semi-supervised GAN model. We compare the two different manifold regularization methods: the normalized and unnormalized stochastic finite difference. We observe that the normalization provides an additional performance boost that grows as the number of labeled examples is reduced (Table A1).

**VAT experimental details**: We used the original VAT implementation made available online by the authors. We kept the same hyperparameters that the authors used for their experiments with 4000 labels.

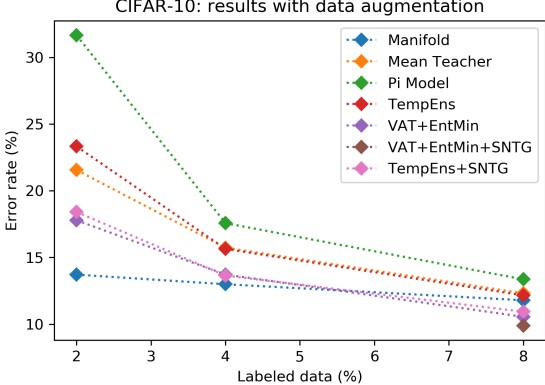

Figure A1: Results including **data augmentation** (4x4 random translations and random horizontal flips similar to previous works).

Table A1: Error rate on CIFAR-10 averaged over 4 runs with different random seeds. Results were obtained without data augmentation.

| CIFAR-10 | 1000 labels (2%) 50000 images | 2000 labels (4%) 50000 images | 4000 labels (8%) 50000 images |
|---|---|---|---|
| Π model | 32.18±1.33 | 23.92±1.03 | 16.55 ± 0.29 |
| Mean Teacher | 30.62 ±1.13 | 23.14 ± 0.46 | 17.74 ± 0.30 |
| VAT (large) | 22.18* | 18.32* | 14.18 |
| VAT+EntMin (large) | 20.52* | 15.52* | 13.15 |
| Π+SNTG | | | 13.62±0.17 |
| VAT+Ent+SNTG | | | **12.49±0.36** |
| Improved GAN | 21.83 ± 2.01 | 19.61 ± 2.09 | 18.63 ± 2.32 |
| Improved GAN + SNTG | | | 14.93 |
| Improved Semi-GAN | 19.52 ±1.5 | | 16.20 ± 1.6 |
| ALI | 19.98 ± 0.89 | 19.09 ± 0.44 | 17.99 ± 1.62 |
| Bad GAN | | | 14.41 ± 0.30 |
| Local GAN | 17.44 ± 0.25 | | 14.23 ± 0.27 |
| Improved GAN (ours) | 17.50 ± 0.34 | 16.80 ± 0.07 | 15.5 ± 0.35 |
| Ambient (ours) | 16.98 ± 0.36 | 15.99 ± 0.14 | 14.75 ± 0.37 |
| Manifold reg. unnormalized (ours) | 18.09 | 15.8 | 14.40 ± 0.21 |
| **Manifold (ours)** | **16.37 ± 0.42** | **15.25 ± 0.35** | 14.34 ± 0.17 |

## B   2D DECOUPLED EXPERIMENTS

**Known manifold:** In Figure A2 we show an example of decoupled training with a perfectly known manifold. We can compute the tangent to each data point (using its polar coordinates:

$\theta = \arctan(\frac{y}{x})$). Instead of the finite difference method, we can plug in the exact tangent direction in our regularizer. As shown in Figure A2 the classifier achieves perfect classification thanks to the tangential directions incorporated to the model.

**Unknown manifold:** In Figure A3 the distribution of the data is unknown. A GAN makes use of the unlabeled examples to learn the data distribution. The data manifold modeled by the GAN is used with by regularizer. It effectively improves generalization power of the model. As shown in Figure A3, all the models achieves perfect classification with only one labeled example per class. An example in the case of a noisy dataset is also given in Figure A11.

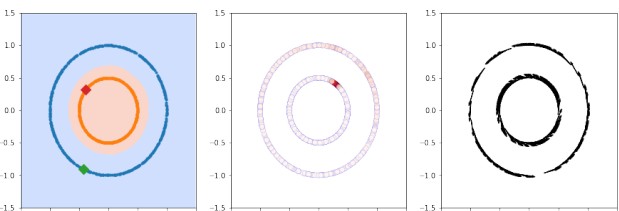

Figure A2: Manifold regularization with known tangent directions on a synthetic dataset. Left: Semi-supervised classification of the disjoint manifold. Middle: Magnitude of the regularization term for a batch of generated samples. Darker fill color reflects larger magnitude. Generated data-points near the decision boundary are highly penalized. Right: Known tangent directions.

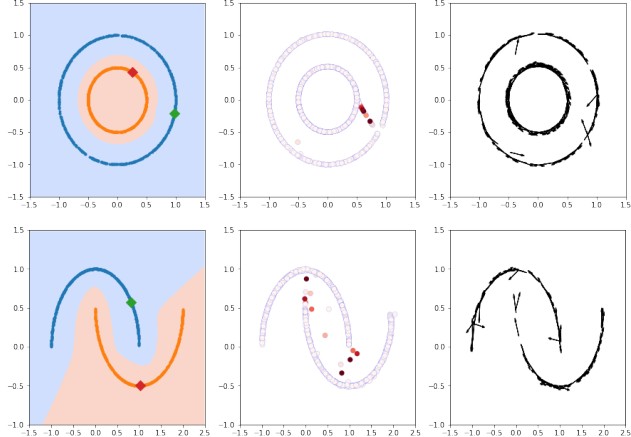

Figure A3: Behavior of manifold regularization with a separate classifier on toy examples. Left: Classification boundaries of trained classifier. Labeled examples are shown as points, unlabeled examples were drawn from distributions over the curves shown. The classifier achieves perfect accuracy on the two datasets. Middle: Magnitude of the regularization term for a batch of generated samples. Darker fill color reflects larger magnitude. Generated data-points near the decision boundary are highly penalized. Right: Direction of invariance promoted by our norm. The trained GANs are able to approximate the data distribution and manifold gradients. In this example $\gamma = 6$ and $\epsilon = 0.15$.

## C   GENERATOR CONDITIONNING

**Log condition number (MLCN):** Following Odena et al. (2018), $G$ defines a mapping from a low dimensional space $Z \in \mathbb{R}^{n_z}$ to the image space $X \in \mathbb{R}^{n_x}$, $G : Z \to X$. For any $z \in Z$ we can define the Jacobian matrix $J_z \in \mathbb{R}^{n_z \times n_x}$. We can write the metric tensor $M_z = J_z^T.J_z$. The condition number of the generator is defined for $M_z$ as $\frac{\lambda_{min}}{\lambda_{max}}$. Large eigenvalues of M correspond to the direction in $Z$ which result in high changes in $G(z)$. A generator with a low mean condition number, should provide smooth changes from $Z$ space to $X$ space.

| CIFAR10 | Mean log condition number |
|---|---|
| Improved GAN | 8.48 |
| Manifold reg (unnormalized) | 7.78 |
| Manifold reg (normalized) | 8.03 |

Table A2: Log condition number of the generator taken at the end of the training. The models are trained with 1000 labels

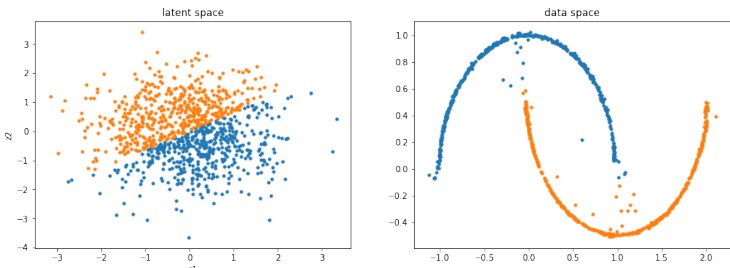

Figure A4: GANs trained on a disjoint manifold. Visualization of the non continuous mapping.

We computed the MLCN by first computing the Jacobian for a set of random samples in the $Z$ space and then we extracted the eigenvalues of the resulted tensors by doing SVD. We report in Table A2 the results of our analysis concerning the conditioning of the generators. It shows that the manifold regularization has a positive effect on the conditioning of the generator.

## D   DISJOINT MANIFOLD LEARNING

We provide an example of two disjoint manifold learnt by GANs in Figure A4. The latent space is a 2D Gaussian vector. This example reveals the issue with generative latent variable models: such models may place points from disjoint manifolds in the data space nearby in the latent space (see left plot).

## E   GENERATED IMAGES

We provide in Figure A6 samples of generated images of the trained generator at the end of the training with and without regularization. Mode collapse is visible for the Improved GAN images with no regularization. While both ambient and manifold regularizations seem to increase images quality.

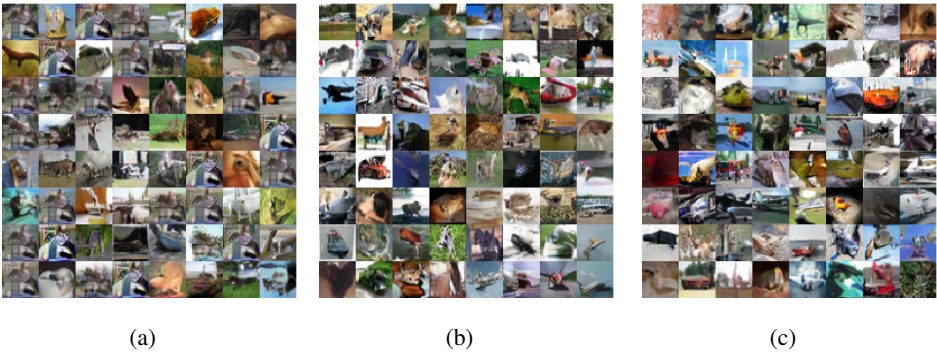

(a)                              (b)                              (c)

Figure A5: Generated images from the coupled experiments trained on CIFAR-10. (a) Improved GAN (b) Ambient regularization (c) Manifold regularization. Mode collapse is visible for the Improved GAN images with no regularization.

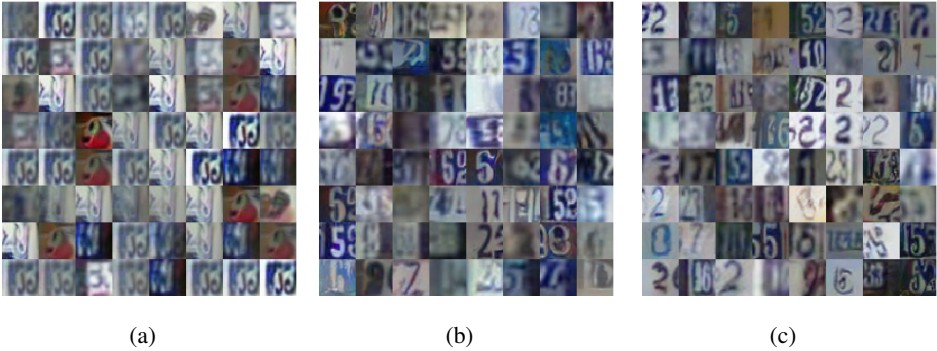

(a)                                      (b)                                      (c)

Figure A6: Generated images from the coupled experiments trained on SVHN. (a) Improved GAN (b) Ambient regularization (c) Manifold regularization. Mode collapse is visible for the Improved GAN images with no regularization.

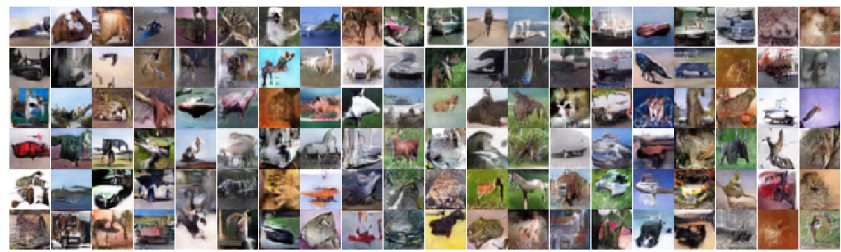

Figure A7: Generated images for decoupled experiments

## F    TANGENT IMAGES

We show samples from the decoupled generator trained on the unlabeled examples on CIFAR10, in Figure A7. The model has an inception score of 6.68 and a FID of 32.32

In this section, for a better understanding of our method, we provide examples of tangent images, approximated by our trained GAN. We shows the influence of the different hyperparameters $\eta, \epsilon$ in Figure A9 and A8 on the visual aspect of the images.

$\epsilon = 0$
$\epsilon = 5$
$\epsilon = 10$
$\epsilon = 20$
$\epsilon = 40$
$\epsilon = 60$

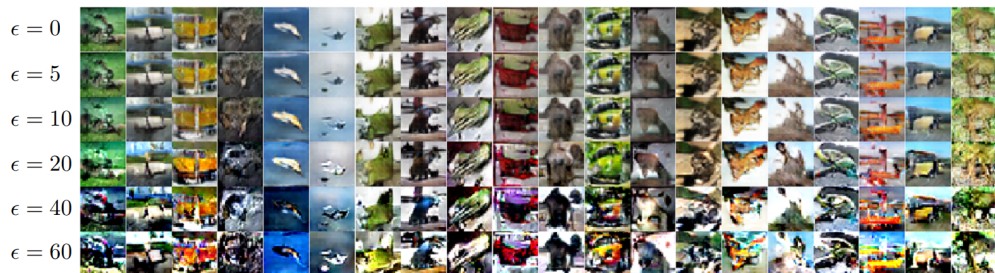

Figure A8: Effect of hyperparameter $\epsilon$ for approximating manifold regularization. Generated images with varying perturbations as per the gradient approximation $(g(z) + \epsilon \bar{r})$ are shown for $\eta = 1$. We used $(\epsilon = 20, \eta = 1)$ in our experiments.

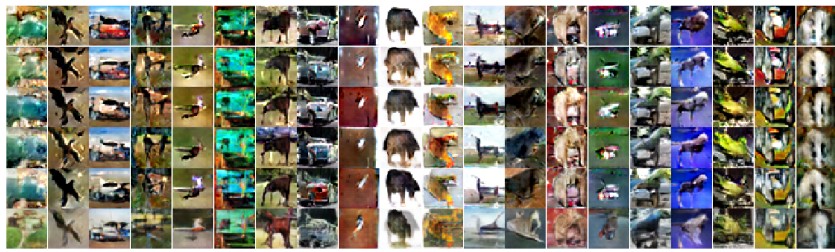

Figure A9: Effect of hyperparameter $\eta$ for approximating manifold regularization. Moving $\eta$ in the range $[10^{-4}, 10^{-3}, 10^{-2}, 10^{-1}, 1, 10]$ with $\epsilon = 20$ fixed

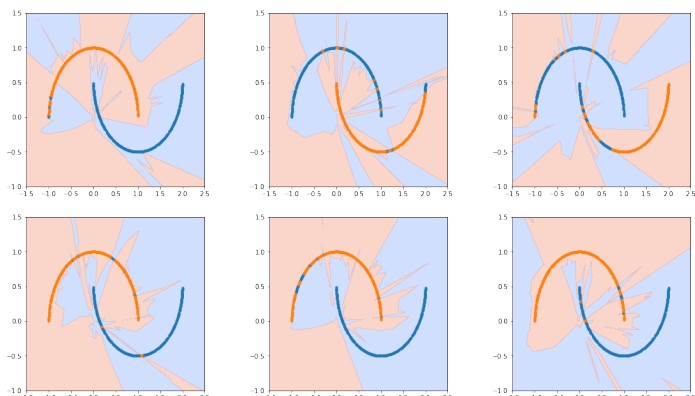

Figure A10: Unsupervised learning of the two moon dataset. No labels are used for the training. The objective function is made of the Laplacian norm, a $l_2$ norm and the entropy of the output prediction $\Omega(f) = \gamma_L \|f\|_L + \gamma_K \|f\|_K^2 - \gamma_h H(f)$. We added the entropy to avoid degenerated solutions. In this example $\epsilon = 0.15$, $\gamma = 3$, $\gamma_K = 1$, $\gamma_h = 0.1$

## G  UNSUPERVISED LEARNING OF SIMPLE MANIFOLDS BY MINIMIZING THE LAPLACIAN NORM

Similarly to Belkin et al. (2006) we show that this is possible to go even further with the manifold regularizer. Under certain conditions, we noticed that it was possible to cluster without any supervision (Figure A10).

## H  SEMI SUPERVISED LEARNING ON A NOISY DATASET

Finally, Figure A11 proves the robustness of the method to the noise. The model successfully performs perfect classification with only one labeled datapoint per class on a noisy Two Moons dataset.

## I  ARCHITECTURES & HYPER PARAMETERS

### I.1  TOY EXAMPLE

For the GAN, we used an simple setup. The generator and discriminator both have 6 ReLU layers of 384 neurons. The latent space of the generator has a latent space of two dimensions. The generator has two outputs, the discriminator has one. The networks are trained under RMSProp. The networks were trained using consensus optimization, introduced in this method is very effective to stabilize the training of the GAN and to successfully capture highly multi modal distributions. The neural network we used for the classification has a similar architecture than the discriminator with 6 ReLU layers of 384 neurons.

Figure A11: Semi-supervised learning of the the noisy two moons dataset after 200,400,800,1000,1200,1400,1600,1800,2000 iterations. We chose for all our 2D experiments the hyper parameters $\epsilon$=0.15, $\gamma = 6$ ,$\eta$=0.01

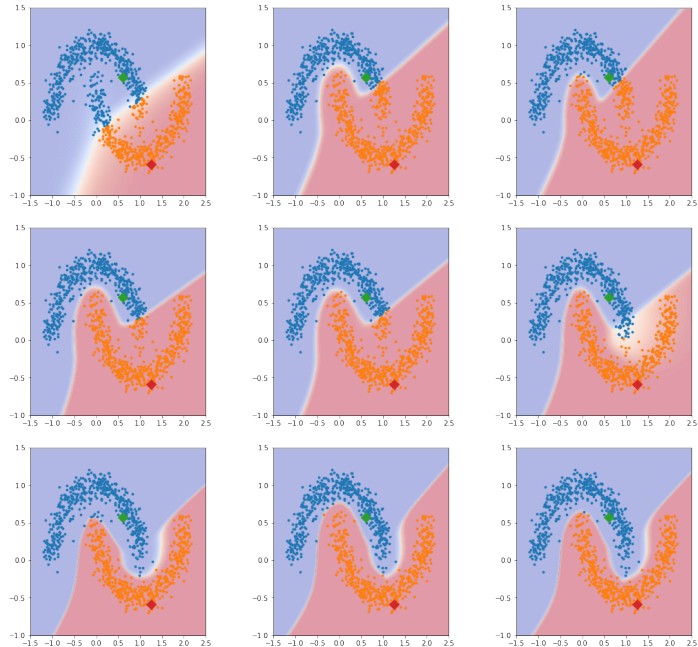

Table A3: Generator architecture we used in our semi supervised GANs experiments

| CIFAR-10 & SVHN |
| --- |
| latent space 100 (uniform noise) |
| dense $4 \times 4 \times 512$ batchnorm ReLU |
| $5\times5$ conv.T stride=2 256 batchnorm ReLU |
| $5\times5$ conv.T stride=2 128 batchnorm ReLU |
| $5\times5$ conv.T stride=2 3 weightnorm tanh |

## I.2 SEMI-SUPERVISED GANS

For reproducibility sake, we provide in this section the exact architectures and hyper-parameters that we used for our different models. We provide in Tables A3 and A4 the architectures of the coupled models. The corresponding hyper-parameters are given in Table A5.

## I.3 DECOUPLED CONVNET

We provide in this section the architectures and hyper-parameters of the models used for the decoupled experiments. In Table A6 we give the architecture of the GAN we trained to learn the distribution of the training examples. Table A7 describes the architecture of the convolutional neural network we used to perform semi supervised classification. Finally, Table A8 shows hyperperameters we found, with a grid-search on the validation set for the training of our models.

Table A4: Discriminator architecture we used in our semi supervised GANs experiments taken from

| conv-large CIFAR-10 | conv-small SVHN |
|---|---|
| 32×32×3 RGB images | |
| dropout, $p = 0.2$ | |
| 3×3 conv. weightnorm 96 lReLU | 3×3 conv. weightnorm 64 lReLU |
| 3×3 conv. weightnorm 96 lReLU | 3×3 conv. weightnorm 64 lReLU |
| 3×3 conv. weightnorm 96 lReLU stride=2 | 3×3 conv. weightnorm 64 lReLU stride=2 |
| dropout, $p = 0.5$ | |
| 3×3 conv. weightnorm 192 lReLU | 3×3 conv. weightnorm 128 lReLU |
| 3×3 conv. weightnorm 192 lReLU | 3×3 conv. weightnorm 128 lReLU |
| 3×3 conv. weightnorm 192 lReLU stride=2 | 3×3 conv. weightnorm 128 lReLU stride=2 |
| dropout, $p = 0.5$ | |
| 3×3 conv. weightnorm 192 lReLU pad=0 | 3×3 conv. weightnorm 128 lReLU pad=0 |
| NiN weightnorm 192 lReLU | NiN weightnorm 128 lReLU |
| NiN weightnorm 192 lReLU | NiN weightnorm 128 lReLU |
| global-pool | |
| dense weightnorm 10 | |

Table A5: Hyperparameters of the models based on the validation set used to report our semi-supervised GAN experiments

| Hyperparameters | CIFAR | SVHN |
|---|---|---|
| $\gamma$ | $10^{-3}$ | $10^{-3}$ |
| $\epsilon$ | 20 | 20 |
| $\eta$ | 1 | 1 |
| Epoch | 1400 | 400 |
| Batch size | 25 | 50 |
| Optimizer | ADAM($\alpha = 3 * 10^{-4}, beta1 = 0.5$) | |
| Learning rate | linearly decayed to 0 after 1200 epochs | no decay |
| Leaky ReLU slope | 0.2 | |
| Weight initialization | Isotropic gaussian ($\mu = 0, \sigma = 0.05$) | |
| Biais initialization | Constant(0) | |

Table A6: GAN architecture we used for our experiments

| Discriminator | Generator |
|---|---|
| 32×32×3 RGB images | latent space 100 (gaussian noise) |
| 4×4 64 conv stride=2 batchnorm lReLU | 4×4 1024 batchnorm conv.T pad=0 ReLU |
| 4×4 256 conv stride=2 batchnorm lReLU | 4×4 256 stride=2 batchnorm conv.T ReLU |
| 4×4 1024 conv stride=2 batchnorm lReLU | 4×4 64 stride=2 batchnorm conv.T ReLU |
| 4×4 1 conv pad=0 | 4×4 3 stride=2 batchnorm conv.T tanh |

Table A7: The convolutional network architecture we used in our decoupled experiments.

| CIFAR-10 & SVHN convnet |
| :---: |
| $32 \times 32 \times 3$ |
| $3 \times 3$, 96 conv. batchnorm lReLU |
| $3 \times 3$, 96 conv. batchnorm lReLU |
| $3 \times 3$, 96 conv. batchnorm lReLU |
| $2 \times 2$ maxpool |
| dropout = 0.5 |
| $3 \times 3$, 128 conv. batchnorm lReLU |
| $3 \times 3$, 128 conv. batchnorm lReLU |
| $3 \times 3$, 128 conv. batchnorm lReLU |
| $2 \times 2$ maxpool |
| dropout=0.5 |
| $3 \times 3$, 256 conv. batchnorm lReLU pad=0 |
| $1 \times 1$, 128 conv. batchnorm lReLU |
| $1 \times 1$, 128 conv. batchnorm lReLU |
| average pooling |
| 10 dense |

Table A8: Hyperparameters of the models based on the validation set used to report our decoupled convnet experiments

| Hyperparameters | CIFAR |
| :--- | :--- |
| $\gamma$ | $10^{-4}$ |
| $\epsilon$ | 20 |
| $\eta$ | 1 |
| Epoch | 200 |
| Batch size | 100 |
| Optimizer | ADAM($\alpha = 3 * 10^{-4}, beta1 = 0.9$) |
| Learning rate | no decay |
| Leaky ReLU slope | 0.2 |
| Weight initialization | Isotropic gaussian ($\mu = 0, \sigma = 0.05$) |
| Biais initialization | Constant(0) |

