# OpenReview forum: "Manifold regularization with GANs for semi-supervised learning"
_ICLR.cc/2019/Conference_

### Official Review · AnonReviewer1 · 2018-11-02
**A simple but interesting idea, but the results are not very significant and some baselines are missing.**

**Rating:** 5
**Confidence:** 4

**Review:**

The paper tackles the problem of semi-supervised classification using GAN-based models. They proposed a manifold regularization by approximating the Laplacian norm using the stochastic finite difference. The motivation is that making the classifier invariant to perturbations along the manifold is more reasonable than random perturbations. The idea is to use GAN to learn the manifold. The difficulty is that (the gradient of) Laplacian norm is impractical to compute for DNNs. They stated that another approximation of the manifold gradient, i.e. adding Gaussian noise \delta to z directly (||f(z) - f(g(z+\delta))||_F) has some drawbacks when the magnitude of noise is too large or too small. The authors proposed another improved gradient approximation by first computing the normalized manifold gradient \bar r(z) and then adding a tunable magnitude of \bar r(z) to g(z), i.e., ||f(z) - f(g(z) +\epsilon \bar r(z) )||_F. Since several previous works Kumar et al. (2017) and Qi et al. (2018) also applied the idea of manifold regularization into GAN, the authors pointed out several advantages of their new regularization.

Pros:
- The paper is clearly written and easy to follow. It gives some intuitive explanations of why their method works.
- The idea is simple and easy to implement based on a standard GAN.
- The authors conduct various experiments to show the interaction of the regularization and the generator.

Cons:
- For semi-supervised classification, the paper did not report the best results in other baselines. E.g., in Table 1 and 2,  the best result of VAT (Miyato et al., 2017) is VAT+Ent, 13.15 for CIFAR-10 (4000 labels) and 4.28 for SVHN (1000 labels). The performance of the proposed method is worse than the previous work but they claimed "state-of-the-art" results. The paper also misses several powerful baselines of semi-supervised learning, e.g. [1,2]. The experimental results are not very convincing because many importance baselines are neglected.
- The paper does not have a significant novel contribution, but rather extends GANs (improved-GAN mostly) with a manifold regularization, which has been explored in many other works Kumar et al. (2017) and Qi et al. (2018).

I'm wondering whether other smoothness regularizations can achieve the same effect when applied to semi-supervised learning, e.g. spectral normalization[3]. It would be better to compare with them.

References:
[1] Adversarial Dropout for Supervised and Semi-Supervised Learning, AAAI 2018
[2] Smooth Neighbors on Teacher Graphs for Semi-supervised Learning, CVPR 2018
[3] Spectral Normalization for Generative Adversarial Networks, ICLR 2018

---

> ### Author Response · Authors · 2018-11-26
> **Thanks for the feedback - we've updated baselines**
>
> Thank you very much for your constructive comments.
>
> First, with respect to baselines, we have updated the results tables to include the additional baselines mentioned, as well as runs for VAT(+EntMin) with lower numbers of labels on CIFAR-10. After updating these baselines, we note that our method still achieves state-of-the-art performance in the regime where 1000 and 2000 labels are used for training on CIFAR-10, with and without data augmentation. We have also updated the text to tone down the claims.
>
> In addition, we note that the highest performance in many of the mentioned baselines (and with VAT) are obtained with a combination of multiple approaches. When our method is compared head-to-head against the proposed method in the mentioned papers, it is competitive and sometimes outperforms them, for instance, in experiments on CIFAR-10 with 4000 labels
>
> With augmentation:
> Adversarial Dropout [1] (11.32) vs ours (11.79 +/- 0.25)
>
> Without augmentation:
> Improved GAN + SNTG [2] (14.93) vs ours (14.34 +/- 0.17)
>
> Defining the best combination of techniques to achieve the highest performance is an interesting direction of future work; our preliminary experiments combining Mean Teacher with manifold regularization have shown some improvements and we will include the results in the final version of the paper.
>
> Second, with respect to novelty, we would like to re-iterate our contributions since they may not have been clear. First, while manifold regularization has been explored in (Kumar et al 2017) and (Qi et al 2018), we proposed an efficient and effective approximation of manifold regularization that is far easier to compute than the involved method in (Kumar et al 2017). Moreover, we point out issues with the standard finite difference approximation to the Jacobian regularization and propose a solution to this problem by ignoring the magnitude of the gradient and using only the direction information. Moreover, we showed manifold regularization provides significant improvements to image quality and linked it to gradient penalties used for stabilizing GAN training, which were not shown by (Qi et al 2018).
>
> We did try to use spectral normalization but did not observe any gains for semi-supervised learning.
>
> Finally we would like to emphasize the conceptual differences between our method and other smoothing methods like spectral normalization - such methods perform isotropic regularization, whilst ours performs anisotropic smoothing along the manifold directions of generated data-points. We showed through experiments using (isotropic) ambient regularization that anisotropic regularization is more beneficial in the case of semi-supervised learning.

---

### Official Review · AnonReviewer3 · 2018-11-02
**Borderline: Manifold Regularization with GANS for SEMI-Supervised Learning**

**Rating:** 5
**Confidence:** 4

**Review:**

This paper builds upon the assumption that GANs successfully approximate the data manifold, and uses this assumption to regularize semi-supervised learning process.
The proposed regularization strategy enforces that a discriminator or a given classifier should be invariant to small perturbations on the data manifold z. It is empirically shown that naively enforcing such a constraint by randomly adding noise to z could lead to under-smoothing or over-smoothing in some cases which can harm the final classification performance. Consequently, the proposed regularization technique takes a step of tunable size in the direction of the manifold gradient, which has the effect of smoothing along the direction of the gradient while ignoring its norm.

Extensive experiments have been conducted, showing that the proposed approach
outperforms or is comparable with recent state-of-the-art approaches on cifar 10, especially in presence of fewer labelled data points. On SVHN however, the proposed approach fails in comparison with (Kumar et al 2017) but performs better than other approaches.

Furthermore, it has been shown that adding the proposed manifold regularization technique to the training of GAN greatly improves the image quality of generated images (in terms of FID scores and inception scores). Also, by combining the proposed regularizer with a classical supervised classifier (via pre-training a GAN and using it for regularization) decreases classification error by 2 to 3%.

Finally, it has also been shown that after training a GAN using the manifold regularization, the algorithm is able to produce similar images giving a low enough perturbation of the data manifold z.

Overall, this paper is well written and show significant improvements especially for image generation. However, the novelty is rather limited as similar ideas have been undertaken (e.g., Mescheder et al 2018), but in different contexts. The paper would be improved if the following points are taken into account:

A comparison with Graph Convolutional Network based techniques seems appropriate (e.g. Kipf and Welling 2017).
How do the FID/Inception improvements compare to (Mescheder et al 2018)?
It would be interesting to discuss why the FID score for SVHN gets worse in presence of 1000 labels.
Although there is a clear improvement in FID scores for Cifar10. It would be informative to show the generated images w/ and w/o manifold regularization.
More analysis should be provided on why (Kumar et al 2017) perform so well on SVHN.
It should be stated that bold values in tables do not represent best results (as it is usually the case) but rather results for the proposed approach.

---

> ### Author Response · Authors · 2018-11-26
> **Thanks for the feedback - we've addressed your questions and clarified novelty of our approach.**
>
> Thank you for your constructive comments. We are glad that you found our experiments extensive and that our approach provides significant improvements.
>
> In response to your comment that "similar ideas have been undertaken (e.g., Mescheder et al 2018), but in different contexts" we would like to take this opportunity to clarify the novelty of our approach.
>
> First, with regards to (Mescheder et al 2018), our method is not simply the application of existing gradient penalties (GPs) in the context of semi-supervised learning. Our approach is conceptually different since the regularizer proposed by (Mescheder et al 2018) is an (isotropic) ambient regularizer in the input space, whereas the regularizer we used performs (anisotropic) smoothing on the manifold parametrized by the latent generative model. We believe we are the first to show the benefits of anisotropic Jacobian regularizers in the context of semi-supervised learning.
>
> Moreover, an important contribution of our work is the efficient computation of such gradient penalties in the context of semi-supervised learning. Current application of such penalties uses the exact Jacobian which is especially computationally expensive in the case of semi-supervised learning as it is now a tensor (one matrix per class in the case of Improved GAN), which quickly becomes intractable with large numbers of classes. We proposed and demonstrated the effectiveness of an efficient (non-obvious) approximation of the Jacobian-based regularizer which significantly accelerates training.
>
> We provide responses to further questions/comments below:
>
> Q: "A comparison with Graph Convolutional Network based techniques seems appropriate (e.g. Kipf and Welling 2017)."
> A: Methods such as (Kipf and Welling 2017) are designed for semi-supervised learning on graphs; here a key challenge is in defining the structure (edges and edge weights) of the graph. Defining the graph structure is not trivial for the image datasets commonly used as benchmarks. In this light, one of the advantages of our approach is that the manifold (graph structure) is implicitly learned by the GAN, thus avoiding the need to explicitly define it. That said, it is an interesting direction for future work and we thank the reviewer for the suggestion.
>
> Q: "How do the FID/Inception improvements compare to (Mescheder et al 2018)?"
> A: We cannot directly compare our image generation scores with those reported in (Merscheder et al 2018) as we used different GAN architectures; for reference, they reported an Inception score of 6.2. We have updated the paper with Inception/FID scores from the ambient regularizer on CIFAR-10 (Table 4), which is an approximation of the proposed regularizer in (Merscheder et al. 2018) using stochastic finite differences. As mentioned earlier, it is not practical to compare the non-approximated regularizer due to the substantial increase in computational complexity in the semi-supervised GAN setting. We observe that ambient regularization gives better image generation scores; however it does not perform as well on semi-supervised learning. This tradeoff between image generation and semi-supervised learning performance was previously reported in (Salimans et al., 2016) "Improved Methods for Training GANs".
>
> Q: "It would be interesting to discuss why the FID score for SVHN gets worse in presence of 1000 labels."
> A: We re-checked our FID computation for this case and fixed a bug. We have updated the paper with updated FID scores; we note there is a high variance in the FID so while there is an improvement on average, it occasionally may not be better.
>
> Q: "Although there is a clear improvement in FID scores for Cifar10. It would be informative to show the generated images w/ and w/o manifold regularization."
> A: We have included generated images with and without manifold regularization in the Appendix (Figure A5 for CIFAR-10, Figure A6 for SVHN) - these show clear improvements as well.
>
> Q: "More analysis should be provided on why (Kumar et al 2017) perform so well on SVHN."
> A: We note that although on average the method of (Kumar et al 2017) performs better on SVHN, the standard deviation is also much higher than many other methods (including ours) on both SVHN and CIFAR-10 indicating that it is not as robust.
>
> Q: "It should be stated that bold values in tables do not represent best results (as it is usually the case) but rather results for the proposed approach."
> A: We have revised the tables such that bold values represent the best results for clarity.

---

### Official Review · AnonReviewer2 · 2018-11-02
**Interesting Approach with Good Results**

**Rating:** 7
**Confidence:** 4

**Review:**

Review for MANIFOLD REGULARIZATION WITH GANS FOR SEMISUPERVISED LEARNING
Summary:
The paper proposed to incorporate a manifold regularization penalty to the GAN to adapt to semi-supervised learning. They approximate this penalty empirically by calculating stochastic finite differences of the generator’s latent variables.
The paper does a good job of motivating the additional regularization penalty and their approximation to it with a series of experiments and intuitive explanations. The experiment results are very through and overall promising. The paper is presented in a clear manner with only minor issues.
Novelty/Significance:
The authors’ add a manifold regularization penalty to GAN discriminator’s loss function. While this is a simple and seemingly obvious approach, it had to be done by someone. Thus while I don’t think their algorithm is super novel, it is significant and thus novel enough. Additionally, the authors’ use of gradients of the generator as an approximation for the manifold penalty is a clever.
Questions/Clarity:
It would be helpful to note in the description of Table 3 what is better (higher/lower). Also Table 3 seems to have standard deviations missing in Supervised DCGANs and Improved GAN for 4000 labels. And is there an explanation on why there isn’t an improvement in the FID score of SVHN for 1000 labels?
What is the first line of Table 4? Is it supposed to be combined with the second? If not, then it is missing results. And is the Pi model missing results or can it not be run on too few labels? If it can’t be run, it would be helpful to state this.
On page 11, “in Figure A2” the first word needs to be capitalized.
In Figure A1, why is there a dark point at one point in the inner circle? What makes the gradient super high there?
What are the differences of the 6 pictures in Figure A7? Iterations?

---

> ### Author Response · Authors · 2018-11-26
> **Thanks for the feedback - we've addressed your questions/comments**
>
> Thank you for your encouraging comments especially with regards to novelty and thoroughness of our experiments.
>
> We have addressed the minor issues you highlighted; answers to your questions are also provided below:
>
> Q: "It would be helpful to note in the description of Table 3 what is better (higher/lower)."
> A: We have updated the table description (now Table 4) as suggested.
>
> Q: "Also Table 3 seems to have standard deviations missing in Supervised DCGANs."
> A: The results we reported were taken from (Gulrajani et al. 2017) which did not include standard deviations for this setting.
>
> Q: "And is there an explanation on why there isn’t an improvement in the FID score of SVHN for 1000 labels?"
> A: We re-checked our FID computation for this case and fixed a bug. We have updated the paper with updated FID scores; we note there is a high variance in the FID so while there is an improvement on average, it occasionally may not be better.
>
> Q: "What is the first line of Table 4? Is it supposed to be combined with the second?"
> A: It is the reference from which all 3 results (Supervised, Pi Model and Mean Teacher) were taken from; we have made this clearer in the revised paper.
>
> Q: "And is the Pi model missing results or can it not be run on too few labels?"
> A: We have updated the table with results on fewer labels reported in (Tarvainen & Valpola, 2017); results on fewer labels were not reported in (Laine & Aila 2017).
>
> Q: "In Figure A1, why is there a dark point at one point in the inner circle? What makes the gradient super high there ?"
> A: This is because the classifier is not smooth in this region. The classifier is probably not constrained enough.
>
> Q:"What are the differences of the 6 pictures in Figure A7? Iterations?"
> A: These are the results from 6 different runs.

---

### Author Response · Authors · 2018-11-26
**Updated baselines, generated images and fixed various issues**

Dear Reviewers,

Thank you for your reviews and constructive feedback. We have included some new figures and updated tables in our paper as a result of the feedback.

Briefly, the changes we made are as follows:
- We updated results tables with additional baselines.
- We moved the table for results including data augmentation into the main text to show a comparison with newer baselines.
- We included generated images with and without manifold regularization or ambient regularization (SVHN and CIFAR-10) in the Appendix (Figure A5).
- Due to the lack of space, we moved the tangent images figure in appendix (Figure A7).
- We have addressed other minor comments.

We would be happy to address any further questions or concerns.

---

### Meta-Review · Area_Chair1 · 2018-12-16
**Borderline paper: reasonably good SSL results but limited novelty**

**Confidence:** 5
**Recommendation:** Reject

**Metareview:**

The paper proposes a method to perform manifold regularization for semi-supervised learning using GANs. Although the SSL results in the paper are competitive with existing methods, R1 and R3 are concerned about the novelty of the work in the light of recent manifold regularization SSL papers with GANs, a point that the AC agrees with. Given the borderline reviews and limited novelty of the core method, the paper just falls short of the acceptance threshold for ICLR.